# Seagrass and Oyster Reef Restoration in Living Shorelines: Effects of Habitat Configuration on Invertebrate Community Assembly

**Cassie M. Pinnell \*, Geana S. Ayala, Melissa V. Patten and Katharyn E. Boyer**

Estuary & Ocean Science Center, San Francisco State University, 3150 Paradise Drive, Tiburon, CA 94920, USA; gsayala@sfsu.edu (G.S.A.); mvpatten@gmail.com (M.V.P.); katboyer@sfsu.edu (K.E.B.)
\* Correspondence: cpinnell@vollmarconsulting.com

**Abstract:** Restoration projects provide a valuable opportunity to experimentally establish foundational habitats in different combinations to test relative effects on community assembly. We evaluated the development of macroinvertebrate communities in response to planting of eelgrass (*Zostera marina*) and construction of reefs intended to support the Olympia oyster (*Ostrea lurida*) in the San Francisco Estuary. Plots of each type, alone or interspersed, were established in 2012 in a pilot living shorelines project, and quarterly invertebrate monitoring was conducted for one year prior to restoration, and three years post-restoration using suction sampling and eelgrass shoot collection. Suction sampling revealed that within one year, oyster reefs supported unique invertebrate assemblages as compared to pre-restoration conditions and controls (unmanipulated mudflat). The eelgrass invertebrate assemblage also shifted, becoming intermediate between reefs and controls. Interspersing both types of habitat structure led eelgrass invertebrate communities to more closely resemble those of oyster reefs alone, though the eelgrass assemblage maintained some distinction (primarily by supporting gammarid and caprellid amphipods). Eelgrass shoot collection documented some additional taxa known to benefit eelgrass growth through consumption of epiphytic algae; however, even after three years, restored eelgrass did not establish an assemblage equivalent to natural beds, as the eelgrass sea hare (*Phyllaplysia taylori*) and eelgrass isopod (*Pentidotea resecata*) remained absent or very rare. We conclude that the restoration of two structurally complex habitat types within tens of meters maximized the variety of invertebrate assemblages supported, but that close interspersion dampened the separately contributed distinctiveness. In addition, management intervention may be needed to overcome the recruitment limitation of species with important roles in maintaining eelgrass habitat.

**Keywords:** eelgrass; oyster; living shoreline; invertebrate; restoration; epifauna; habitat structure

---

## 1. Introduction

The study of habitat structure includes assessing the impacts of biotic and abiotic arrangements on ecological patterns and processes [1,2]. The complexity of these structural arrangements has been investigated as a driving factor in species diversity and abundance across terrestrial and aquatic habitat types and is often referred to using a range of terms including habitat heterogeneity, habitat complexity, and spatial heterogeneity [2,3]. That biodiversity may be enhanced through structurally complex habitat configurations is highly relevant to conservation management and restoration programs [4–6].

The degree of habitat complexity is often determined by foundation species, which provide the physical structure on which other species depend. Foundation species engineer their environment by strongly influencing the availability of resources to other organisms through habitat creation, modification, or maintenance [7]. The loss of these organisms can, therefore, have heavy consequences for entire ecosystems, including associated biota [8].

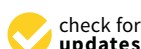



The restoration of foundation species provides a unique opportunity to observe the extent of habitat creation, modification, or maintenance, including the re-introduction of habitat structure.

Within estuaries, seagrass and shellfish beds serve as foundational habitats by providing structure for a multitude of ecologically important invertebrate and fish species [9,10]. These structurally complex habitats have been shown to support greater densities of decapod crustaceans than bare marsh-edge habitats [11,12] and higher densities of fish and invertebrates than unstructured mudflat [13]. Seagrass beds serve as a large food source through both direct (grazing) and indirect means (epiphyte and invertebrate habitat) [14]. Seagrass and shellfish (including oysters, clams, and mussels) act as ecosystem engineers by contributing to water quality by reducing wave action and thus turbidity, increasing sediment accretion [15–18], and providing nutrient cycling and carbon storage [10,19].

Due to a number of largely human-mediated factors, ecologically important seagrass and shellfish habitats are declining worldwide. For example, many populations of the Olympia oyster (*Ostrea lurida*), the only native oyster on the west coast of the United States, were depleted by the 1930s due to extensive harvesting, pollution, poor water quality, and potentially the impacts of introduced species [20]. The extent of native oysters elsewhere (*Crassostrea virginica* and *Ostrea conchaphila* in North America, *O. edulis* in Europe, and *O. anagasi* and *Saccostrea glomerata* in Australia) are currently estimated at less than 10% of their historical abundance due to overfishing and habitat destruction [21]. Eelgrass (*Zostera marina*) populations and their associated biotic communities are sensitive to a wide array of anthropogenic disturbances, including mechanical impacts from shipping and dredging [22,23] and water quality issues resulting from nutrient inputs or increases in turbidity [24]. Additionally, rising sea levels and changing estuarine conditions (including reduced accretion rates) threaten eelgrass populations [25] and could result in additional eelgrass losses worldwide in the future. Due to their ecological value, several species of native oysters as well as eelgrass are targeted for restoration throughout the world [14].

The San Francisco (SF) Estuary is the largest Pacific estuary in North and South America, the second largest estuary in the United States, and its watershed covers approximately 40% of California [26]. The SF Estuary provides habitat to hundreds of species of fish, aquatic invertebrates, mammals and birds, including multiple endangered species [27,28]. In addition to providing extensive habitat, the SF Estuary provides all four ecosystem services (provisioning, regulating, cultural, and supporting) identified by the Millennium Ecosystem Assessment [29] to the over 7 million local residents [30]. Intertidal eelgrass, oysters, and their associated aquatic invertebrate communities provide ecosystem services, including food (provisioning); water quality improvements (regulating); significance of oysters to native people (cultural); and nutrient cycling (supporting).

In 2010, the San Francisco Bay Subtidal Habitat Goals Project proposed an increase of 3200 hectares of both eelgrass and native oysters within the next 50 years [24]. In addition to the general benefits of seagrass and shellfish, eelgrass and native oysters in the SF Estuary provide specific, local benefits. One of California's most economically valuable aquatic invertebrates, the Dungeness crab (*Metacarcinus magister*) [28], relies heavily on eelgrass and oyster habitats for juvenile shelter and food sources [31–35]. Additionally, the use of these species as living shorelines to mitigate wave impacts and resulting flooding related to sea level rise is currently being locally investigated in a number of projects spearheaded by the California State Coastal Conservancy.

The first of these living shoreline projects was implemented in 2012, and we endeavored to assess habitat provided through the addition of living three-dimensional structure. Specifically, we sought to assess the efficacy of restored intertidal eelgrass beds and native Olympia oyster reefs alone and together in colonization and use by invertebrates over the first several years following installation relative to pre-installation conditions and un-manipulated control plots. We hypothesized that oyster reef and eelgrass habitats would attract different assemblages of species and that together the diversity of taxa would be substantially greater than in either habitat alone due to the additive design. Further-

more, we investigated the degree to which invertebrate assemblages in restored eelgrass beds mirrored those in natural beds to better understand the timescale of establishing the invertebrate community following restoration. The findings from this study can be used to inform the design of future intertidal restoration projects and efforts to maximize supported species and their associated functions.

## 2. Materials and Methods

### 2.1. Location

The study site ("San Rafael" or "study site") was in San Rafael, CA, north of the Richmond–San Rafael Bridge in Marin County (Figure 1). This site was selected for the California State Coastal Conservancy's living shorelines ("Near-shore Linkages") pilot project in part because the site appeared well suited to support eelgrass and native oysters. Depth ($-0.6$ m MLLW) was appropriate for both species, small test plantings of eelgrass had persisted for five years, and oysters were present on riprap along the shoreline. This area was also identified as suitable for eelgrass by a biophysical model [36]. The substrate was predominantly clay-silt and firm enough to support oyster reefs. The Nature Conservancy was a supportive landowner, the site was large enough that the treatment array could be placed parallel to the shore and roughly perpendicular to the dominant wave forces, and the distance from shore (~200 m) was close enough for monitoring access on foot. Records do not exist at the scale needed to determine whether this study site historically supported either species. However, due to the highly altered environment of the SF Estuary (dredging, filling, shoreline modification, and extensive sediment deposition), the site was selected based on current suitability rather than recorded historical observations.

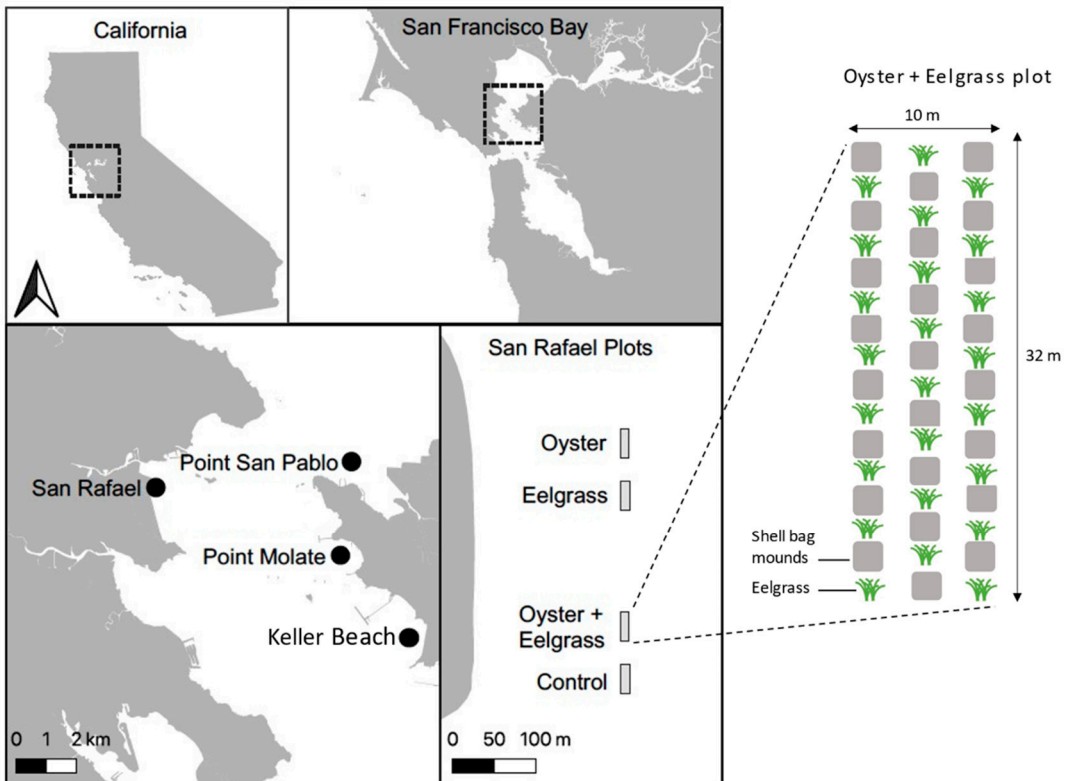

**Figure 1.** Map of study site location (San Rafael, Marin County, California, 'SR'; 122°29′15″ W, 37°57′47″ N), as well as layout of treatment plots. Transplanted eelgrass was sourced from natural beds across the bay at Point San Pablo ('PSP'; 122°25′02″ W, 37°58′06″ N) and Point Molate ('PM'; 122°24′47″ W, 37°56′36″ N). In addition to samples collected from the study site, eelgrass shoot samples were collected at natural beds at Keller Beach ('KB'; 37°55′15″ N, 122°23′12 W) and Point Molate to compare invertebrate assemblages.

Eelgrass epifaunal communities developing at the San Rafael living shorelines project were compared to those of natural eelgrass beds at Keller Beach and Point Molate, across the bay along the Richmond shoreline (Figure 1). This stretch of shoreline holds the closest natural beds to the study site; hence, these beds presented the best opportunity to sample invertebrate communities that are representative of local dispersal events and comparable regional conditions within the central bay of the SF Estuary.

## 2.2. Treatment Design

Six pre-treatment plots (spaced approximately 66 m apart) were established along a 330 m sampling transect to span the length of the future restoration treatment plots. After pre-treatment surveys were completed, treatment plots were installed. The four 10 × 32 m treatment plots were (Figure 1): (1) 'Oyster' ('O'), which consisted of mounded bags of Pacific oyster shell with a footprint of 1 m × 1 m per element. Elements were installed in groups of four to make larger square units of 4 m$^2$. Three rows of eight units were installed, for a total of 24 units in the treatment plot (96 elements). To minimize scour, square spaces of the same size (4 m$^2$) were included between these oyster reef units; (2) 'Eelgrass' ('E') was planted with the same 4 m$^2$ spacing as the oyster reef units. Eelgrass units measured 1.5 m × 1.5 m; therefore, the central 1.5 m × 1.5 m (2.25 m$^2$) space within every other 4 m$^2$ space was planted. Eelgrass was sourced from natural beds across the estuary at Point San Pablo and Point Molate. Whole shoots were attached to bamboo stakes [36] and planted at a density of 24 shoots per unit; (3) oyster/eelgrass combination plot included oyster ('EO(O)') and eelgrass ('EO(E)') elements that were combined using an additive design, with eelgrass placed into the central 2.25 m$^2$ of the 4 m$^2$ spaces between oyster units; (4) 'Control' ('C') plot of the same size consisted of only bare mudflat. The four treatments were assigned randomly with a minimum of 30 m between each treatment plot; a gap was left in the middle of the array to avoid test plots of eelgrass that were established in 2008 to test the suitability of the site (Figure 1).

## 2.3. Monitoring

Pre-treatment ('P') surveys were conducted quarterly from October 2011 through July 2012 (four surveys). Oyster elements were installed in July 2012 and eelgrass planting followed in August 2012. These late-season plantings did not survive, and the plots were replanted in spring 2013. Post-restoration monitoring occurred six times between August 2013 and August 2015 (summer 2013, fall 2013, winter (January) 2014, spring 2014, summer 2014, and summer 2015).

Monitoring included two methods: 1) suction sampling, a method that could be used consistently across all treatments and 2) shoot collection, a common method used for sampling invertebrates on eelgrass [37,38], which permitted the assessment of invertebrate assemblages with and without oyster reefs present and comparisons with natural eelgrass beds.

### 2.3.1. Suction Sampling

Suction sampling employed a hand-held, battery-operated aquarium vacuum (Super Battery Vac, Penn-Plax Inc., Hauppauge, NY, USA) with a modified opening of approximately 10 mm to sample the epibenthic aquatic invertebrates in one 0.25 × 0.25 m quadrat at each sampling point. Suction samples were collected from the vertical structure (eelgrass shoots or oyster reef in E, O, and EO plots) or the water column (C plot or pre-treatment) and separately from the benthos (either between the eelgrass plants, at the base of the oyster reef, or on the mudflat of the control plots or pre-treatment), sampling for 30 s each for vertical structure and substrate. Six suction samples were collected across the future project area during each of the four pre-treatment surveys, combining the water column and benthos sample at each location. During the first four post-treatment surveys (August 2013–April 2014), six samples (combined benthos with water column or structure) each were collected from the E, O, and C plots, and from the eelgrass (EO(E)) and oyster (EO(O))

units within the combination plot (12 samples total in the EO plot). During the last two post-treatment surveys (August 2014 and August 2015), three samples were collected from each of the treatments. Sampling was conducted at tidal heights when water depths were between 0.3 m and 1.0 m at the treatment plots. Sampling positions within the plots were selected using a random number generator.

A layer of fine nylon mesh (pantyhose) was connected to the output of the vacuum, allowing all water to pass through while trapping fine sediments and invertebrates. The mesh was then removed and placed in ethanol to preserve the sample in the field. In the lab, the sample was washed through a 500 $\mu$m sieve to separate the sediment from the invertebrates. If the sample volume was greater than 10 mL, it was split to $\frac{1}{2}$ using a Motoda Plankton Sample Splitter (Aquatic Research Instruments, Hope, ID). All samples were stored in 20 mL glass scintillation vials in 70% ethanol until they could be sorted. Samples were then stained using rose Bengal dye, and invertebrates were sorted to the lowest possible taxonomic level and counted under a dissecting microscope.

### 2.3.2. Eelgrass Shoot Sampling

Whole vegetative shoots were collected to compare epiphytic invertebrates on eelgrass with (EO(E)) and without (E) oyster reefs present, as follows: (1) year one—39 EO(E) and 32 E shoots collected in summer 2013, 16 EO(E) and 29 E in fall 2013, 29 EO(E) and 29 E in spring 2014; (2) year two—31 EO(E) and 30 E in summer 2014, 22 EO(E) and 22 E in fall 2014, 15 EO(E) and 32 E in spring 2015; (3) year three—20 EO(E) and 18 E in summer 2015.

To compare the establishment of invertebrates at the restored sites to assemblages in natural sites, 10 vegetative shoots each were collected at Keller Beach ('KB') and Point Molate ('PM') in fall 2013 (PM only), spring 2014, fall 2014, spring 2015, and summer 2015. Shoots were separated from the rhizome at the sediment surface and placed into plastic bags for transport to the lab. Once in the lab, shoots were rinsed with freshwater through a 500 $\mu$m sieve to separate invertebrates from the shoots [36]. Invertebrate samples were stored in 70% ethanol, stained with rose Bengal dye, then sorted and enumerated (per shoot) under a dissecting microscope at 10× power.

### 2.4. Data Analysis

All data analyses were performed using R (version 3.4.1). As the assumptions for parametric statistics were not met (Shapiro test for normality and Bartlett test for homogeneity of variances), and since each large treatment plot was sub-sampled (therefore, not true replicates), abundance and species richness were compared between treatments using Kruskal–Wallis rank sum tests (pgirmess package) and non-parametric post hoc multiple comparison procedures, including Tukey honesty significant difference tests. Ecological community data were visualized using correspondence analysis (CA) in the ade4 package in R [39] and differences in species assemblages were assessed using permutational multivariate analysis of variance (perMANOVA) based on the Bray–Curtis dissimilarity metric using the adonis implementation in the vegan package for R [40].

### 3. Results

A total of 38 invertebrate taxa were identified and enumerated for this study (Appendix A Tables A1–A3). Suction sampling included 23 taxa across mudflat, water column, eelgrass, and oyster reefs at the San Rafael living shorelines site, before and three years after installation. Eelgrass shoot sampling included 33 taxa at the San Rafael site and two natural eelgrass beds, Keller Beach and Point Molate over several years of sampling.

### 3.1. Eelgrass and Oyster Reef Habitat Development

#### 3.1.1. Taxa Richness and Abundance: Suction Sampling

Suction sampling showed taxa richness increased significantly between the pre-treatment and post-treatment period for all structured treatments ($X^2 = 38.72$, df = 5, $p < 0.001$); however, the control was intermediate and not different from pre-treatment or

the structured treatment plots (Tukey HSD tests) due to recruitment of new taxa across the project site. No significant difference was observed in overall invertebrate abundance between pre-treatment and experimental period data ($X^2$ = 9.85, df = 5, $p$ = 0.08) (Appendix A Figures A1 and A2).

### 3.1.2. Community Assemblage Development: Suction Sampling

After one year post-installation, correspondence analyses showed a significant difference between invertebrate community assemblages by treatment and season, and in a treatment × season interaction (PerMANOVA $p < 0.001$ for all three; Appendix A Table A4). After one year, invertebrate assemblages with oyster reefs present were similar with or without eelgrass present (O and EO(O)) (Figure 2a). The eelgrass (E) invertebrate community was intermediate between the control/pre-treatment and oyster/combination communities, sharing some taxa with both (Figure 2a). Notably, the assemblage associated with eelgrass shifted markedly toward that of oyster reefs when the latter was present nearby, but not vice versa. Over the course of three years, this pattern remained, but with an increased overlap between the eelgrass and oyster reef communities (Figure 2b).

(**a**)

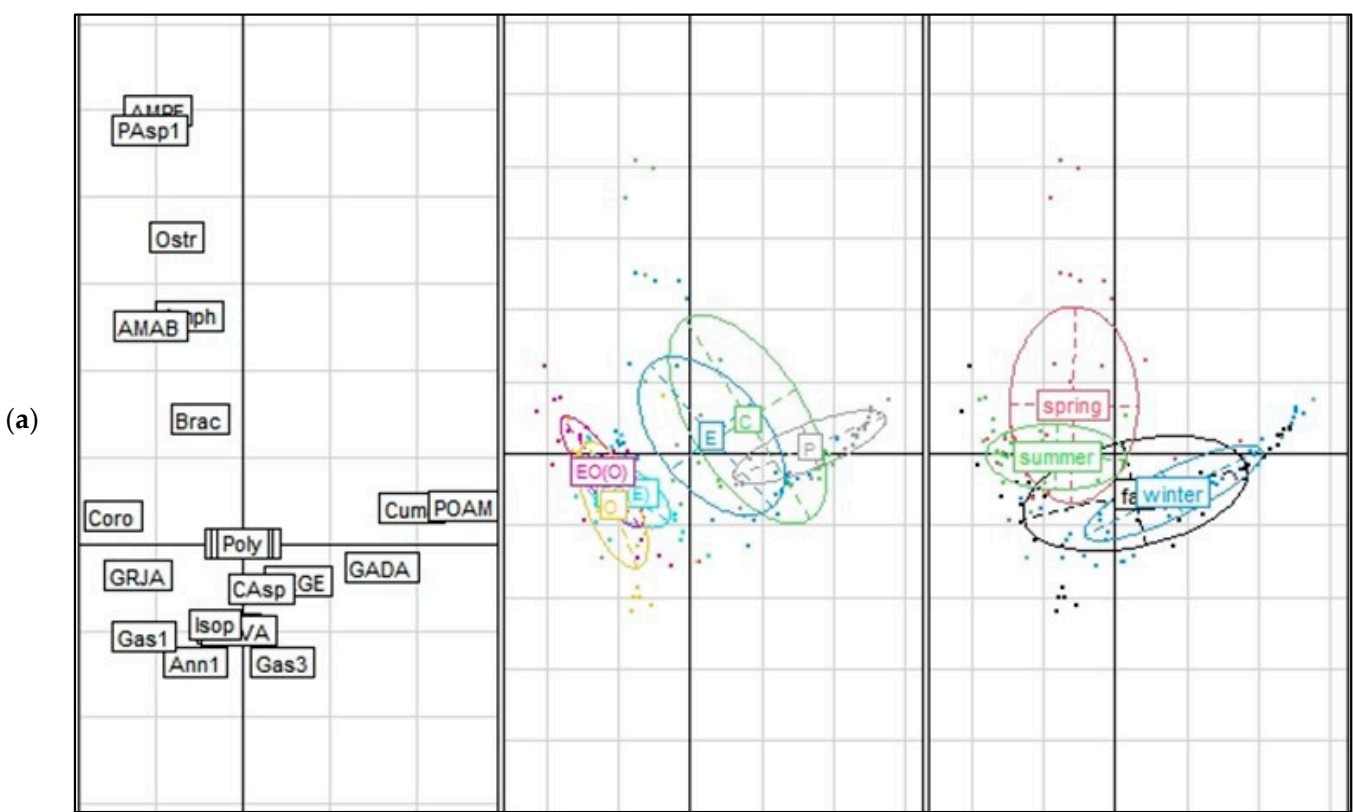

**Figure 2.** *Cont.*

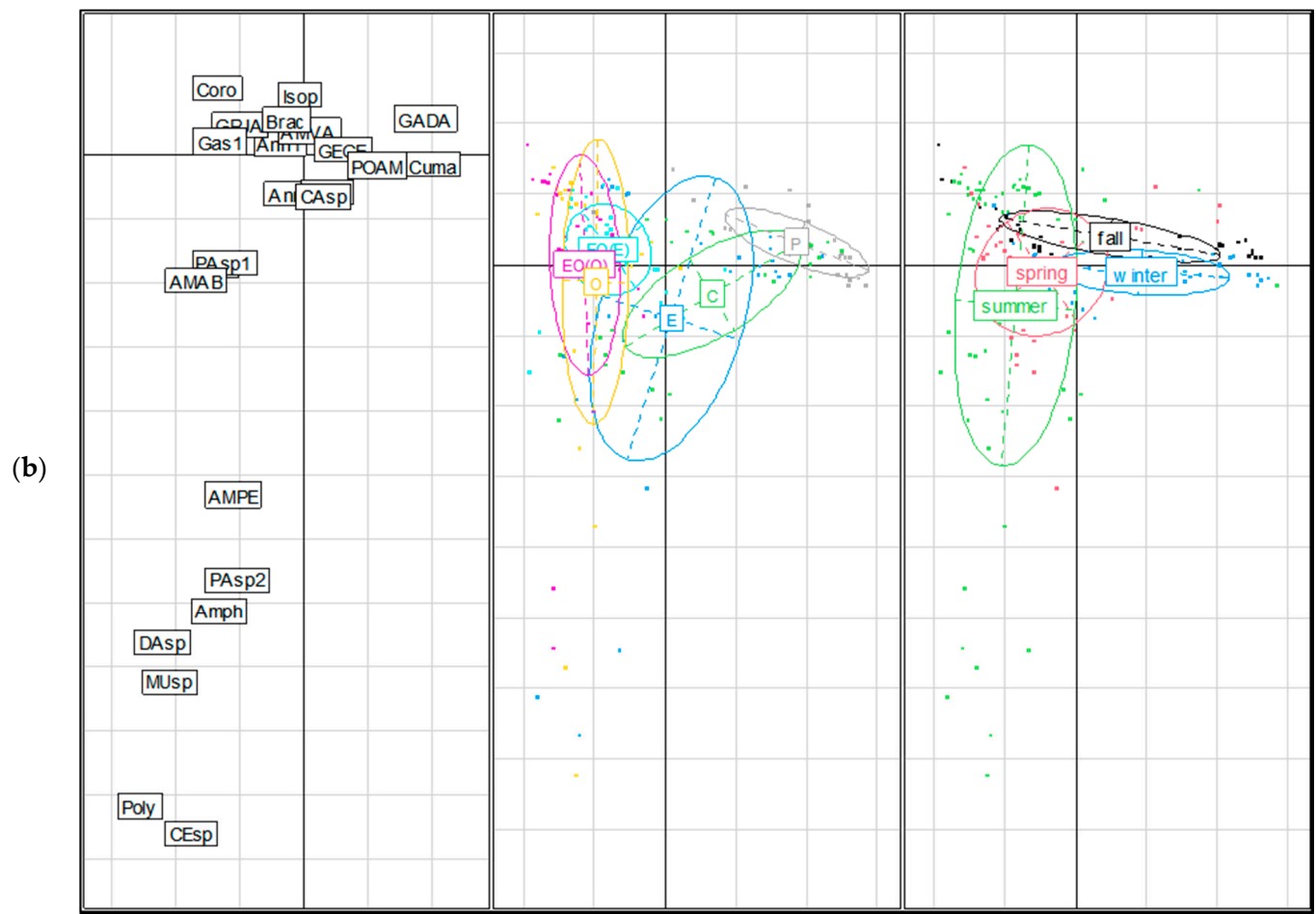

**Figure 2.** (**a**) Correspondence analysis of taxa detected by suction sampling during pre-treatment (Pre, Oct 2011–July 2012) and first year after treatment (July 2013–April 2014); and (**b**) Correspondence analysis of taxa detected by suction sampling during pre-treatment (Pre, Oct 2011–July 2012) and first year after treatment (July 2013–April 2014), second year after treatment (summer only, July 2014), and third year after treatment (summer only, July 2015). Panels show taxa (**left**) coded by treatment (**center**) and season (**right**), with all taxa. Treatment abbreviations as in previous figures, taxa abbreviations detailed in Appendix A Tables A1 and A2. Correspondence Analysis (ade4) Factors 1 and 2: (**a**) Inertia = 38.18%; and (**b**) Inertia = 28.19%.

In Figure 2b, the taxa displayed in the right upper quadrant corresponded most with pre-treatment and control communities, including *Potamocorbula amurensis* [POAM], *Gammarus daiberi* [GADA], and Cumacea [Cuma]. The far-left quadrants, associated with the oyster and combination assemblages, included Corophiidae (both *Monocorophium* sp. and *Corophium* sp.), *Ampelisca abdita* [AMAB], *Grandidierella japonica* [GRJA], and an unidentified snail [Gas1]. The strongest driver of difference between the eelgrass communities (E and EO(E)) and non-eelgrass communities (P, C, O, and EO(O)) was *Caprella* sp. [CAsp]. Additional taxa associated with the eelgrass communities were the non-native *Ampithoe valida* [AMVA], Brachyura [Brac; crab megalopae], and Annelida [Ann1, Ann2; from the sediment at the base of the eelgrass]. These trends were noticeable after the first year of treatment (Figure 2a), and continued through all three years (Figure 2b).

There were strong similarities among the fall and winter communities, and among the summer and spring communities (Figure 2a,b). The significant treatment × season interaction points to the structure-associated species being more prevalent in the spring and summer months. Accordingly, we explored summer-only patterns, as we had the most sampling points during this season over time. Analyses of summer communities illustrate an evolving invertebrate assemblage, with each year progressively developing

away from pre-treatment (year 'Zero') conditions (Figure 3). The community assemblage observed one year after treatment installation (year 'One' = 2013) was most similar to that of pre-treatment ('Zero'), with the following years ('Two' = 2014 and 'Three' = 2015) each shifting further from pre-treatment.

Taxa with the largest impacts on the differentiation of community assemblage between summers include Amphipoda [Amph] and *Cerapus* sp. [CEsp] in year three (Figure 3). Additional taxa affecting the differentiation in the developing communities include *A. abdita* and *P. amurensis* in year two, and *Munna* sp. [MUsp], Polyplacophora [Poly], *Paranthura* sp. [PAsp2], *Daphnia* sp. [DAsp], and *Americhelidium pectinatum* [AMPE] in year three. With all summer periods of monitoring included, the community assemblages by treatment were similar to those observed after the first year of monitoring, with stronger similarities between the control (C) and eelgrass (E), and continued close association between the treatments that included oyster reefs (O, EO(E), and EO(O)).

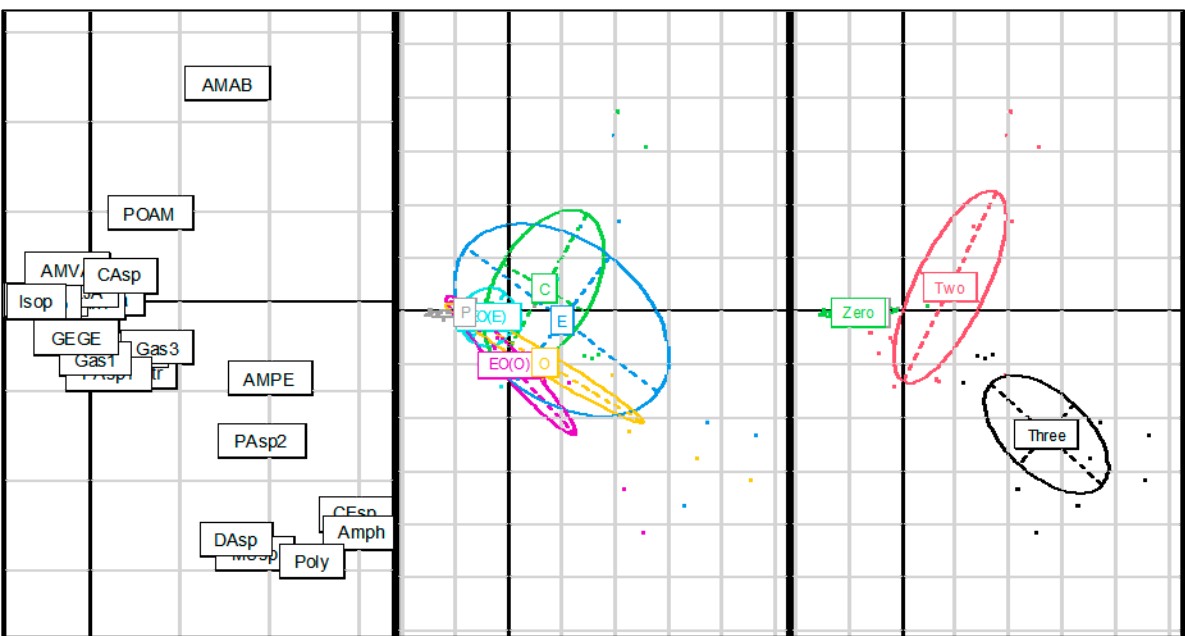

**Figure 3.** Correspondence analysis of taxa detected across four years of summer season monitoring, including pre-treatment ('Zero' July 2012), and after treatment ('One' July 2013 (shown behind 'Zero'), 'Two' August 2014, and 'Three' August 2015) suction sampling. Panels show taxa (**left**) coded by treatment (**center**) and year (**right**), with all taxa. Treatment abbreviations as in previous figures, taxa abbreviations detailed in Appendix A Tables A1 and A2. Correspondence Analysis (ade4) Factors 1 and 2, Inertia = 39.82%.

### 3.2. Comparison of Eelgrass Epifaunal Restored and Natural Communities

3.2.1. Taxa Richness and Abundance: Shoot Collections

Significantly higher invertebrate taxa richness was observed on the shoots of restored eelgrass (E) and combination (EO) shoots from the San Rafael project site ('SR'), as compared to shoots from reference natural eelgrass beds just across the bay at Keller Beach ('KB') and Point Molate ('PM') ($X^2 = 53.85$, df = 3, $p < 0.001$). No significant difference in taxa richness was detected between the natural sites, nor between the treatments (E and EO) at SR (Appendix A Figure A3). Invertebrate taxa abundance on the restored shoots (SR(E) and SR(EO)) was only significantly higher than at one of the reference sites, Point Molate ($X^2 = 47.26$, df = 3, $p < 0.001$). However, the abundance on shoots from Keller Beach was also significantly higher than at Point Molate, and no difference was observed between Keller Beach and restored shoots. No significant difference in abundance was detected between the restored treatments (E and EO) at San Rafael (Appendix A Figure A3).

### 3.2.2. Community Assemblage of Restored and Natural Eelgrass: Shoot Collections

The community assemblage of epifaunal invertebrates collected from eelgrass shoot samples differed significantly by treatment (eelgrass (E), eelgrass combination (EO), and natural (N)), as well as by collection site (PerMANOVA $p < 0.001$) (Appendix A Table A5). The two main drivers of the difference between sites were *Phyllaplysia taylori* [PHTA] and *Pentidotea resecata* [PERE], which were more closely associated with the natural populations (specifically Point Molate), than the restored populations (Figure 4a). Both of these native species are associated with eelgrass; however, only one *P. resecata* individual was ever observed in the restored beds (fall 2014) and *P. taylori* was observed only in very low abundances (mean of 0.1 or 0.2 per shoot at most) in the restored beds.

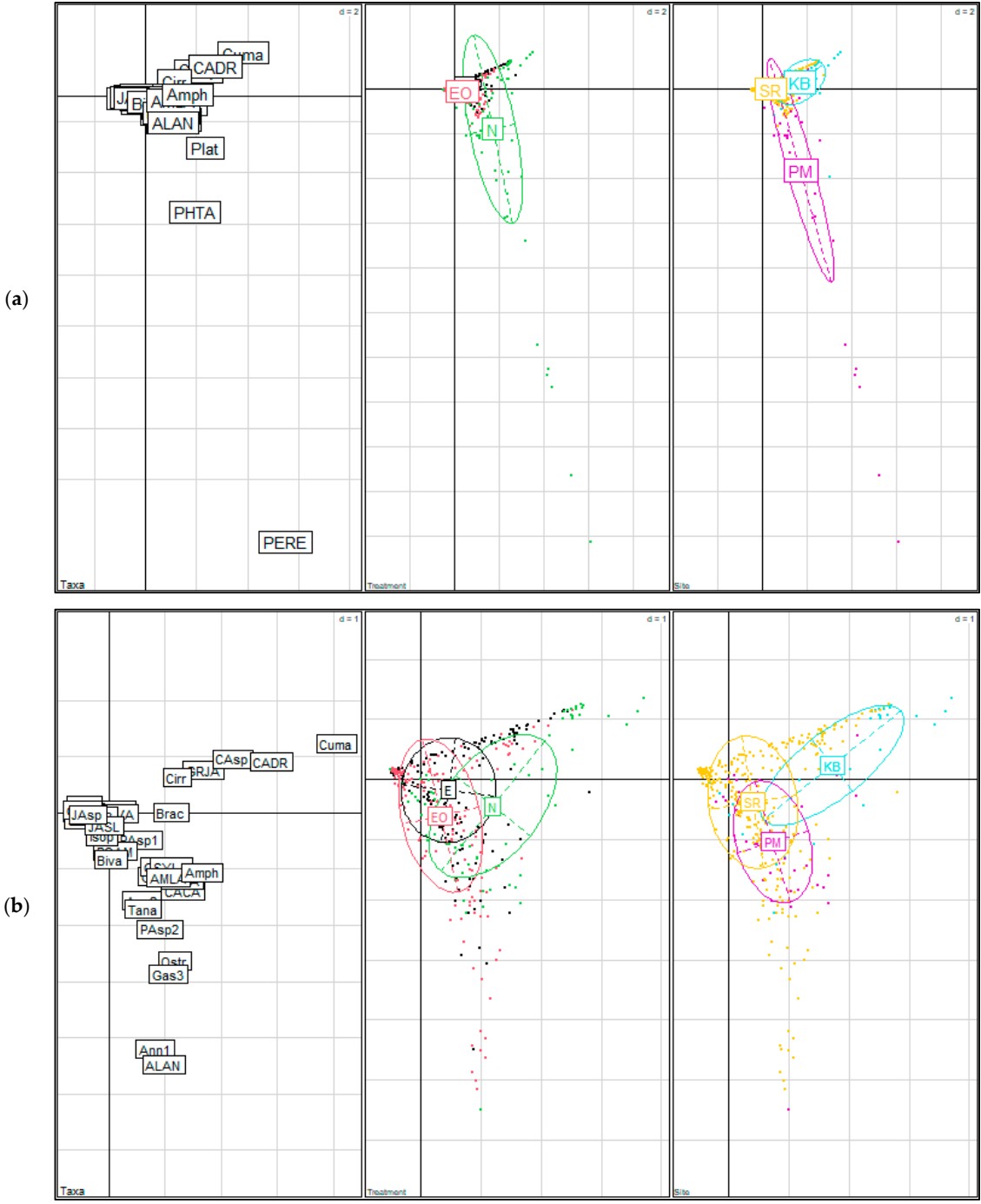

**Figure 4.** Correspondence analysis of taxa detected from shoot samples from restored eelgrass within eelgrass only (E) and

eelgrass and oyster combination plots (EO), and natural (N) eelgrass populations. Samples collected at natural sites Keller Beach (KB, n = 36) and Point Molate (PM, n = 41), and restored populations at the San Rafael project site (SR, n = 365) across three years of post-treatment sampling. Panels show taxa (**left**) coded by treatment (**center**) and site (**right**), with (**a**) all taxa included: Adonis (Bray–Curtis) *p* < 0.001; Correspondence Analysis (ade4) Factors 1 and 2, Inertia 21.93% and (**b**) *Pentidotea resecata* [PERE], *Phyllaplysia taylori* [PHTA], and Platyhelminthes [Plat] removed to spread results. Adonis (Bray–Curtis) *p* < 0.001; Correspondence Analysis (ade4) Factors 1 and 2, Inertia 26.69%. Taxa abbreviations detailed in Appendix A Table A3.

By removing *P. resecata*, *P. taylori*, and Platyhelminthes [Plat] from the correspondence analysis, we are able to better visualize the influence of less influential taxa on specific assemblages (Figure 4b). Taxa such as Cumacea and *Caprella drepanochir* [CADR] were more prevalent in the natural (N) eelgrass beds, specifically Keller Beach, whereas taxa such as *Allorchestes angusta* [ALAN], Annelida, and *Paranthura* sp. were predominantly detected in the restored communities (E) and (EO) at San Rafael. Restored populations (E) and (EO) were somewhat differentiated from each other in addition to varying from the natural populations, with a stronger association of *Caprella* sp. and Brachyura megalopae in eelgrass-only plots. In addition to differences between the restored and natural sites, the Keller Beach and the Point Molate natural beds were distinct from each other, due to key taxa such as *C. drepanochir*, as referenced above. The differentiation between natural and restored communities, as well as between individual sites, remained consistent between sampling years, with no apparent trend towards community convergence over time (Figure 5). Of all three years, the Year 2-restored community supported a small amount of *P. resecata* and *P. taylori*, and most resembled the natural community at Point Molate.

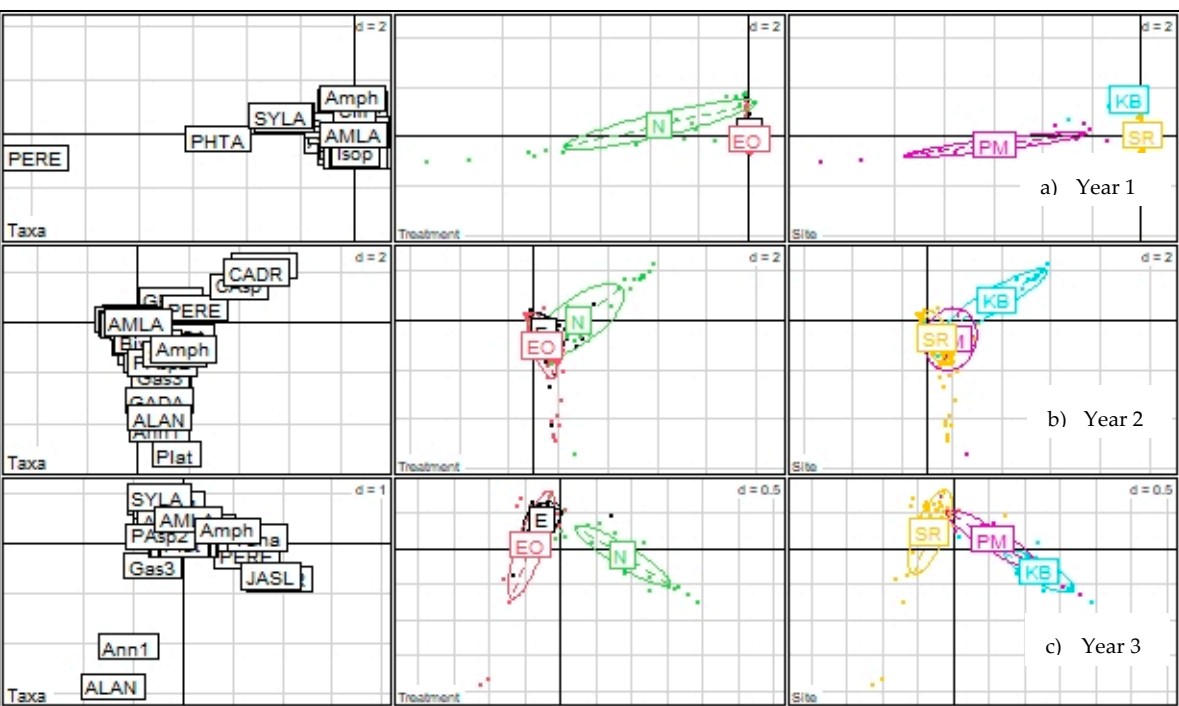

**Figure 5.** Correspondence analysis of taxa detected over the course of three monitoring years from shoot samples from restored eelgrass within eelgrass only (E) and eelgrass and oyster combination plots (EO) from San Rafael (SR), and natural (N) eelgrass populations from Keller Beach (KB) and Point Molate (PM). Panels show taxa (**left**) coded by treatment (**center**) and site (**right**), with (**a**) Year 1 monitoring results (Factors 1 and 2, Inertia = 38.24%), (**b**) Year 2 monitoring results (Factors 1 and 2, Inertia = 29.64%), and (**c**) Year 3 monitoring results (Factors 1 and 2, Inertia = 30.89%). Treatment abbreviations as in previous figures; taxa abbreviations detailed in Appendix A Table A3.

## 4. Discussion

The pilot scale restoration of oyster reefs and eelgrass in the first living shorelines project in San Francisco Estuary permitted the opportunity to compare invertebrate assemblage establishment on newly installed three-dimensional structure in comparison to pre-project mudflat conditions and un-manipulated control areas. We hypothesized that oyster reefs and eelgrass would attract different assemblages of taxa and thus that restoring the two together would maximize invertebrate taxa diversity due to the additive design. Although we did not detect a difference in taxa richness on a per suction sample basis, there were persistent differences in community composition between the two habitat types that indicate greater taxa and assemblage variety was supported by having both habitats present at the restoration site. However, interspersion of the habitats shifted the eelgrass assemblage to greatly resemble the oyster assemblage; together, these results may argue for maintaining these habitats nearby (within tens of meters) but without interspersion at a fine scale (meters).

Within the first year of treatment, eelgrass and oyster structures shifted in invertebrate composition as compared to pre-treatment and control plots at the project site. This shift in assemblage supports the idea that restored eelgrass and oyster habitats can quickly begin to support a community that was not prevalent in less complex mudflat habitat [10,11]. Results are also consistent with previous studies that found epibenthic invertebrate densities were significantly higher on oyster and eelgrass substrates than mudflat, and composition was also significantly related to habitat [13].

Additionally, eelgrass alone developed a community assemblage mostly distinct from treatments with oyster reef present, suggesting that community composition is driven in part by eelgrass or oysters specifically, not simply the addition of structure. This difference was largely driven by caprellid amphipods, deposit feeders that cling to eelgrass blades, and by an introduced gammarid amphipod, *Ampithoe valida*, which consumes both epiphytic algae and eelgrass [41,42].

Community assemblage data from eelgrass shoot samples showed that the restored eelgrass communities did not resemble those in natural eelgrass beds, nor do the data suggest a trend of convergence after three years of monitoring. *Pentidotea resecata* and *Phllaplysia taylori*, both eelgrass endemics, were not detected in abundances comparable to those of the natural eelgrass beds during the sampling period of this study. Both increase eelgrass growth by removing epiphytic algae, and *A. valida* damage to eelgrass may be mediated by chemical defenses induced in the plant by *P. resecata* grazing on epiphytes on the leaf surfaces [43]. These native species are both direct developers, which could limit their rates of establishment in the restored beds [44]; interestingly, *A. valida* shares this trait but recruited early, suggesting it may more readily ride on detached eelgrass flowering shoots or vegetative wrack to new locations.

Shoot samples collected from the natural beds at Keller Beach and Point Molate exhibited invertebrate assemblages largely different from each other in addition to the restored plots. Invertebrate communities within natural eelgrass beds in the SF Estuary vary by location, as shown by previous studies, perhaps relating to sediment or water conditions [36]. Sediment analyses following our study showed somewhat greater sand and lower organic matter content at Keller Beach than at Point Molate (87 vs. 72% sand and 2.1 vs. 2.4% organic matter, respectively), while the San Rafael site had much lower sand (25%) and higher organic matter (6.2%) than either natural site [45]. We acknowledge that these differences may be important to the invertebrate assemblage that develops at the restored eelgrass bed; however, the extent to which sediment conditions drive composition and abundance relative to other factors, such as dispersal ability, recruitment order, species interactions, and predation, is unclear. Experimentally assisting in the migration of valued native invertebrates, such as *P. resecata* and P. taylori, to the restoration site would be one way to test whether dispersal limitation, versus other factors, influences the presence and persistence of these species.

Within the heavily invaded SF Estuary, restored eelgrass provides a habitat for early colonizing invaders, such as *A. valida*, and eventual colonization by other non-native species not detected during pre-treatment surveys, regardless of conducting freshwater dips to limit initial presence. This prevalence suggests that not only should precautions be followed when sourcing from eelgrass populations with known infestations of introduced taxa, but additional management actions should be considered such as inoculation with preferred native taxa early in a restoration project. However, it is important to note that both native and introduced species provide a valuable food source for larger resident and pelagic organisms. The restoration of eelgrass and oysters can provide a habitat for decapods, including a nursery habitat for the commercially important Dungeness crab (*Metacarinus magister*), and higher numbers of crab larvae and crab megalopae (Brachyura) were observed at the project site following restoration [46]. Further, sampling with baited traps showed that Pacific rock crabs (*Romalean antennarium*) increased in abundance with oyster reef presence as did shrimp species with both eelgrass and reef presence [46].

## 5. Conclusions

We conclude that restoring both eelgrass and oyster reefs together can maximize the diversity of invertebrate assemblages at restoration sites, despite the fact that restored eelgrass invertebrate assemblages did not converge with those of natural beds over the course of the study. Our study design allowed us to develop a nuanced view of how each of these foundational habitats influence the assemblage found in the other; as oyster reef presence in close proximity to eelgrass led to a shift in the composition of the eelgrass invertebrate community, we might maximize the variety of invertebrate assemblages supported by having both habitats present at a restoration site but not closely interspersed. The findings from this study can be used to inform future intertidal restoration projects in the region, and more generally suggest there is value in using restoration sites as an opportunity to better understand community assembly and the relative and interactive effects of the habitat types restored.

**Author Contributions:** Conceptualization, K.E.B., methodology, K.E.B. and C.M.P.; formal analysis, C.M.P.; investigation, C.M.P., M.V.P. and G.S.A.; data curation, C.M.P., M.V.P. and G.S.A.; writing—original draft preparation, C.M.P.; writing—review and editing, K.E.B. and G.S.A.; visualization, C.M.P. and G.S.A.; supervision, K.E.B.; project administration, K.E.B.; funding acquisition, K.E.B. All authors have read and agreed to the published version of the manuscript.

**Funding:** This research was funded by the California State Coastal Conservancy, through grants from the Environmental Protection Agency—SF Bay Water Quality Improvement Fund (#EM-00T34101-0), California State Coastal Conservancy Proposition 84—Safe Drinking Water, Water Quality and Supply, Flood Control, River and Coastal Protection Bond Act (#10-030), and the Golden Gate Bridge Highway and Transportation District, Cooperative Agreement with California State Coastal Conservancy (Sausalito Ferry Terminal Mitigation).

**Institutional Review Board Statement:** Not applicable.

**Informed Consent Statement:** Not applicable.

**Data Availability Statement:** The data presented in this study are available in Appendix A.

**Acknowledgments:** We are grateful to Marilyn Latta for extensive work on planning, fund-raising, implementing, and managing the living shorelines project assessed here. Field and laboratory assistance was provided by Jennifer Miller, Jessica Craft, Lauren Scheinberg, Stephanie Kiriakopolos, Crystal Weaver, Lubomira Raykova, Kevin Stockmann, Rosa Schneider, Evyan Borgnis, and Whitney Thornton. Margot Buchbinder and Julien Moderan provided support on statistics and data visualization. Lina Ceballos provided assistance with taxa identification.

**Conflicts of Interest:** The authors declare no conflict of interest. The funders had no role in the design of the study; in the collection, analyses, or interpretation of data; in the writing of the manuscript, or in the decision to publish the results.

## Appendix A

**Table A1.** Suction-sampled invertebrate community composition and mean abundance per sample in the pre-treatment sampling periods. Letters represent family and order (c = caprellid, g = gammarid, I = isopod), and native (n), introduced (i), or cryptogenic (c). See Figure 1 for site abbreviations.

| Pre-Treatment | |
|---|---|
| Taxa | SR |
| Fall 2011 | |
| Annelida 2 (Ann2) | 0.3 |
| Corophiidae (Coro) | 1.2 |
| Cumacea (Cuma) | 144.5 |
| *Gammarus daiberi* (GADA) (g, i) | 73.0 |
| Gastropoda 3 (Gas3) | 0.7 |
| *Gemma gemma* (GEGE) | 0.7 |
| *Grandidierella japonica* (GRJA) (g, i) | 11.5 |
| Isopoda (Isop) (I) | 0.7 |
| *Potamocorbula amurensis* (POAM) (i) | 0.8 |
| Winter 2012 | |
| Annelida 1 (Ann1) | 2.0 |
| Annelida 2 (Ann2) | 22.8 |
| Corophiidae (Coro) | 2.7 |
| Cumacea (Cuma) | 312.8 |
| *Gammarus daiberi* (GADA) (g, i) | 41.3 |
| Spring 2012 | |
| Annelida 1 (Ann1) | 48.8 |
| Annelida 2 (Ann2) | 13.0 |
| *Caprella* sp. (CAsp) (c) | 2.2 |
| Corophiidae (Coro) | 39.7 |
| Cumacea (Cuma) | 45.8 |
| *Gammarus daiberi* (GADA) (g, i) | 70.0 |
| *Grandidierella japonica* (GRJA) (g, i) | 1.0 |
| Ostracoda (Ostr) | 0.7 |
| *Paradexamine* sp. (PAsp1) (g) | 0.2 |
| Summer 2012 | |
| Annelida 1 (Ann1) | 0.3 |
| Corophiidae (Coro) | 2.0 |
| Cumacea (Cuma) | 0.2 |
| *Gammarus daiberi* (GADA) (g, i) | 2.5 |
| *Grandidierella japonica* (GRJA) (g, i) | 0.2 |
| *Paradexamine* sp. (PAsp1) (g) | 0.2 |

**Table A2.** Suction-sampled invertebrate community composition and mean abundance per sample in the post-treatment sampling periods. Letters represent family and order (c = caprellid, g = gammarid, I = isopod), and native (n), introduced (i), or cryptogenic (c). See Figure 1 for site abbreviations.

| Taxa | SREO (E) | SREO (O) | SR E | SR O | SR C |
|---|---|---|---|---|---|
| Summer 2013 | | | | | |
| *Ampithoe valida* (AMVA) (g, i) | 0 | 0.7 | 0 | 6.7 | 0 |
| Annelida 1 (Ann1) | 5.0 | 7.7 | 32.5 | 41.3 | 15.3 |
| Annelida 2 (Ann2) | 3.7 | 9.7 | 12.8 | 18.0 | 7.0 |
| *Caprella* sp. (CAsp) (c) | 0.3 | 0 | 0 | 0.2 | 0.5 |
| Corophiidae (Coro) | 72.0 | 145.5 | 247.5 | 156.3 | 10.7 |
| Cumacea (Cuma) | 14.5 | 4.3 | 54.5 | 5.3 | 34.8 |
| *Gammarus daiberi* (GADA) (g, i) | 9.3 | 4.8 | 38.3 | 9.8 | 30.8 |
| Gastropoda 3 (Gas3) | 0 | 0 | 4.8 | 0 | 0.3 |
| *Grandidierella japonica* (GRJA) (g, i) | 66.7 | 47.8 | 31.3 | 78.0 | 70.0 |

**Table A2.** *Cont.*

| Taxa | SREO (E) | SREO (O) | SR E | SR O | SR C |
|---|---|---|---|---|---|
| Isopoda (Isop) (I) | 0.2 | 0 | 0 | 0.7 | 0.2 |
| Ostracoda (Ostr) | 0.2 | 0 | 0.2 | 0 | 0.2 |
| *Paradexamine* sp. (PAsp1) (g) | 0.3 | 8.2 | 0 | 6.2 | 0 |
| *Potamocorbula amurensis* (POAM) (i) | 0 | 0 | 0.2 | 0 | 0 |
| Fall 2013 | | | | | |
| *Ampithoe valida* (AMVA) (g, i) | 3.2 | 0 | 11.7 | 0 | 0 |
| Annelida 1 (Ann1) | 14.3 | 3.5 | 27.8 | 110.3 | 11.3 |
| Annelida 2 (Ann2) | 10.0 | 1.3 | 6.2 | 6.7 | 1.7 |
| *Caprella* sp. (CAsp) (c) | 3.3 | 0.2 | 0 | 0 | 0 |
| Corophiidae (Coro) | 11.0 | 11.0 | 0.3 | 0.3 | 0 |
| Cumacea (Cuma) | 2.8 | 5.0 | 41.7 | 4.7 | 71.7 |
| *Gammarus daiberi* (GADA) (g, i) | 0.7 | 0 | 27.3 | 1.3 | 65.8 |
| Gastropoda 1 (Gas1) | 0 | 0.2 | 0 | 0 | 0 |
| Gastropoda 3 (Gas3) | 0 | 0 | 16.7 | 3.7 | 3.3 |
| *Gemma gemma* (GEGE) (i) | 0 | 0 | 0 | 0.2 | 0 |
| *Grandidierella japonica* (GRJA) (g, i) | 59.3 | 36.7 | 14.5 | 24.2 | 12.0 |
| Isopoda (Isop) (i) | 0 | 0 | 0.2 | 0.7 | 0 |
| Ostracoda (Ostr) | 0.2 | 0 | 0 | 0.2 | 0.3 |
| *Paradexamine* sp. (PAsp1) (g) | 0.5 | 0.2 | 0 | 0 | 0 |
| Winter 2014 | | | | | |
| Amphipoda (Amph) | 0 | 0 | 0 | 0.8 | 0 |
| *Ampithoe valida* (AMVA) (g, i) | 5.8 | 2.5 | 0 | 0 | 2.5 |
| Annelida 1 (Ann1) | 13.8 | 17.7 | 7.5 | 47.6 | 4.0 |
| Annelida 2 (Ann2) | 42.8 | 67.2 | 2.5 | 21.8 | 22.5 |
| *Caprella* sp. (CAsp) (c) | 19.2 | 0.5 | 10.7 | 0 | 0 |
| Corophiidae (Coro) | 11.2 | 20.7 | 3.7 | 35.0 | 1.3 |
| Cumacea (Cuma) | 18.0 | 8.7 | 56.8 | 17.2 | 56.2 |
| *Gammarus daiberi* (GADA) (g, i) | 5.3 | 0.5 | 25.0 | 2.8 | 28.5 |
| Gastropoda 1 (Gas1) | 0 | 0.3 | 0 | 0.8 | 0 |
| Gastropoda 3 (Gas3) | 8.5 | 6.8 | 13.3 | 11.6 | 0.2 |
| *Gemma gemma* (GEGE) (i) | 0.3 | 0 | 0 | 0.4 | 0 |
| *Grandidierella japonica* (GRJA) (g, i) | 6.2 | 10.3 | 6.0 | 30.2 | 4.0 |
| Isopoda (Isop) (I) | 0 | 0 | 0.3 | 1.2 | 0 |
| Ostracoda (Ostr) | 0.3 | 0 | 0.7 | 0.2 | 0 |
| Spring 2014 | | | | | |
| *Americhelidium pectinatum* (AMPE) (g, n) | 0.3 | 2.3 | 1.7 | 0 | 13.5 |
| *Ampelisca abdita* (AMAB) (g, i) | 8.8 | 5.5 | 8.8 | 0 | 32.8 |
| Amphipoda (Amph) | 0 | 0 | 0.5 | 0 | 0.5 |
| *Ampithoe valida* (AMVA) (g, i) | 0 | 0 | 1.3 | 0.8 | 0.2 |
| Annelida 1 (Ann1) | 32.2 | 4.2 | 8.5 | 6.5 | 31.8 |
| Annelida 2 (Ann2) | 3.3 | 2.7 | 3.2 | 17.3 | 8.0 |
| Brachyura (Brac) | 0 | 0 | 0 | 0.2 | 0 |
| *Caprella* sp. (CAsp) (c) | 0.2 | 0 | 1.8 | 0.7 | 5.3 |
| Corophiidae (Coro) | 23.5 | 11.5 | 7.2 | 53.8 | 54.3 |
| Cumacea (Cuma) | 5.7 | 10.8 | 18.0 | 5.5 | 56.8 |
| *Gammarus daiberi* (GADA) (g, i) | 0 | 0.3 | 0 | 7.8 | 0 |
| Gastropoda 1 (Gas1) | 0 | 0 | 0 | 0.3 | 0 |
| Gastropoda 3 (Gas3) | 6.7 | 1.2 | 0.8 | 0 | 2.7 |
| *Gemma gemma* (GEGE) (i) | 0 | 0 | 0.2 | 0.2 | 0 |
| *Grandidierella japonica* (GRJA) (g, i) | 47.5 | 68.2 | 15.7 | 27.0 | 10.8 |
| Ostracoda (Ostr) | 2.0 | 0.5 | 1.2 | 0.3 | 11.2 |
| *Paradexamine* sp. (PAsp1) (g) | 1.2 | 0.3 | 14.7 | 6.0 | 63.8 |
| Summer 2014 | | | | | |
| *Americhelidium pectinatum* (AMPE) (g, n) | 0 | 0.3 | 15.3 | 16.0 | 7.7 |
| *Ampelisca abdita* (AMAB) (g, i) | 0.7 | 0.3 | 57.7 | 1.7 | 99.0 |
| *Ampithoe valida* (AMVA) (g, i) | 0 | 0.3 | 3.0 | 0.3 | 0 |
| Annelida 1 (Ann1) | 0.7 | 3.7 | 12.0 | 7.7 | 10.7 |
| Annelida 2 (Ann2) | 0.3 | 16.3 | 15.7 | 12.3 | 0.3 |

**Table A2.** *Cont.*

| Taxa | SREO (E) | SREO (O) | SR E | SR O | SR C |
|---|---|---|---|---|---|
| *Caprella* sp. (CAsp) (c) | 0 | 0 | 0 | 0 | 0.7 |
| Corophiidae (Coro) | 2.7 | 24.3 | 28.3 | 92.3 | 29.0 |
| Cumacea (Cuma) | 0 | 0 | 12.7 | 0.3 | 20.7 |
| *Gammarus daiberi* (GADA) (g, i) | 0.3 | 0.3 | 1.3 | 0 | 0 |
| Gastropoda 1 (Gas1) | 0 | 1.3 | 0 | 0.3 | 0 |
| Gastropoda 3 (Gas3) | 0 | 3.3 | 3.7 | 5.7 | 0 |
| *Gemma gemma* (GEGE) (i) | 0 | 0.3 | 0 | 0 | 0 |
| *Grandidierella japonica* (GRJA) (g, i) | 1.3 | 18.0 | 25.0 | 6.3 | 43.7 |
| Ostracoda (Ostr) | 0 | 4.0 | 2.7 | 14.3 | 0 |
| *Paradexamine* sp. (PAsp1) (g) | 4.7 | 13.7 | 0.3 | 4.0 | 0 |
| *Potamocorbula amurensis* (POAM) (i) | 0 | 0 | 1.3 | 0 | 0 |
| Summer 2015 | | | | | |
| *Americhelidium pectinatum* (AMPE) (g, n) | - | 2.0 | 36.0 | 22.3 | 52.3 |
| Amphipoda (Amph) | - | 0.7 | 0.3 | 3.0 | 0 |
| Annelida 1 (Ann1) | - | 4.3 | 0.7 | 0 | 38.7 |
| Annelida 2 (Ann2) | - | 20.7 | 5.0 | 9.3 | 44.3 |
| *Caprella* sp. (CAsp) (c) | - | 0.3 | 0 | 0 | 0 |
| *Cerapus* sp. (CEsp) (g) | - | 0.7 | 0 | 8.3 | 0.3 |
| Corophiidae (Coro) | - | 9.3 | 1.3 | 5.3 | 0.3 |
| Cumacea (Cuma) | - | 1.7 | 5.7 | 5.3 | 67.7 |
| *Daphnia* sp. (DAsp) | - | 2.7 | 0 | 0 | 0 |
| Gastropoda 3 (Gas3) | - | 0.7 | 2.7 | 3.0 | 1.3 |
| *Grandidierella japonica* (GRJA) (g, i) | - | 6.7 | 1.7 | 6.7 | 54.7 |
| *Munna* sp. (Musp) | - | 16.0 | 1.0 | 2.0 | 0 |
| Ostracoda (Ostr) | - | 1.7 | 2.0 | 3.7 | 0.7 |
| *Paradexamine* sp. (PAsp1) (g) | - | 7.0 | 3.3 | 3.0 | 0 |
| *Paranthura* sp. (PAsp2) (i) | - | 2.3 | 1.3 | 2.0 | 4.0 |
| Polyplacophora (Poly) | - | 0 | 0.3 | 0 | 0 |

**Table A3.** Invertebrate community composition and mean abundance per collected shoot at all sites and years. Letters represent family and order (c = caprellid, g = gammarid, I = isopod), and native (n), introduced (i), or cryptogenic (c). See Figure 1 for site abbreviations.

| Taxa | KB Natural | PM Natural | SR Eelgrass | SR Eelgrass + Oyster |
|---|---|---|---|---|
| Year 1 | | | | |
| Summer 2013 | | | | |
| *Ampelisca abdita* (AMAB) (g, i) | - | - | 7.3 | 0.2 |
| *Ampithoe valida* (AMVA) (g, i) | - | - | 11.7 | 17.6 |
| Annelida 1 (Ann1) | - | - | 0.8 | 3.9 |
| Annelida 2 (Ann2) | - | - | 1 | 2.3 |
| Brachyura (Brac) | - | - | 0.1 | 0 |
| *Caprella* sp. (CAsp) (c) | - | - | 1.3 | 1.3 |
| Cirripedia (Cirr) | - | - | 0.5 | 0.3 |
| Corophiidae (Coro) (g) | - | - | 218.7 | 129.3 |
| Cumacea (Cuma) | - | - | 0.1 | 0.1 |
| *Gammarus daiberi* (GADA) (g, i) | - | - | 0.3 | 1.2 |
| Gastropoda 2 (Gas2) | - | - | 1.1 | 0.1 |
| Gastropoda 3 (Gas3) | - | - | <0.1 | 0.1 |
| *Grandidierella japonica* (GRJA) (g, i) | - | - | 0.3 | 1.8 |
| Isopoda (Isop) (I) | - | - | <0.1 | <0.1 |
| Ostracoda (Ostr) | - | - | 0 | 0.3 |
| *Paradexamine* sp. (PAsp1) (g) | - | - | 14 | 4.1 |
| *Phyllaplysia taylori* (PHTA) (n) | - | - | 0 | 0.1 |
| Amphipoda (Amph) | - | 0.7 | 1.1 | 4.1 |

**Table A3.** *Cont.*

| Taxa | KB Natural | PM Natural | SR Eelgrass | SR Eelgrass + Oyster |
|---|---|---|---|---|
| *Ampithoe valida* (AMVA) (g, i) | - | 12.3 | 19.4 | 25.9 |
| Annelida 1 (Ann1) | - | 0 | 0 | 0.2 |
| Annelida 2 (Ann2) | - | 0.7 | 0.2 | 1.6 |
| *Caprella californica* (CACA) (c, n) | - | 0 | <0.1 | 0.2 |
| *Caprella drepanochir* (CADR) (c, i) | - | 1.7 | 0 | 0 |
| *Caprella* sp. (CAsp) (c) | - | 0 | 46.3 | 39.4 |
| Cirripedia (Cirr) | - | 0 | 13.7 | 14.3 |
| Corophiidae (Coro) (g) | - | 0 | 24.2 | 12.9 |
| Cumacea (Cuma) | - | 0 | 0 | 0.1 |
| *Gammarus daiberi* (GADA) (g, i) | - | 0 | 0 | 0.2 |
| *Grandidierella japonica* (GRJA) (g, i) | - | 0 | 2.4 | 11.4 |
| *Jassa slatteryi* (JASL) (g, c) | - | 9.3 | 0 | 0 |
| Ostracoda (Ostr) | - | 1 | 0 | 0.1 |
| *Paradexamine* sp. (PAsp1) (g) | - | 0 | 2.2 | 8.9 |
| *Pentidotea resecata* (PERE) (I, n) | - | 1 | 0 | 0 |
| *Phyllaplysia taylori* (PHTA) (n) | - | 3 | 0 | 0 |
| *Synidotea laticauda* (SYLA) (I, i) | - | 0.3 | 0 | 0 |
| Spring 2014 | | | | |
| Amphipoda (Amph) | 1.6 | 0 | 0.1 | 0 |
| *Ampithoe valida* (AMVA) (g, i) | 0.9 | 0 | 2.1 | 0.7 |
| Annelida 1 (Ann1) | 0.2 | 0.8 | 0.2 | 0.8 |
| Annelida 2 (Ann2) | 0.5 | 3.5 | 0.6 | 0.7 |
| Brachyura (Brac) | 0.2 | 0.2 | <0.1 | 0 |
| *Caprella* sp. (CAsp) (c) | 128.5 | 0.4 | 48 | 2.5 |
| Cirripedia (Cirr) | 0 | 0 | 9.5 | 0.9 |
| Corophiidae (Coro) (g) | 0.4 | 0.5 | 90.9 | 17.3 |
| *Gammarus daiberi* (GADA) (g, i) | 0 | 0.2 | 0.6 | 2.2 |
| Gastropoda 1 (Gas1) | 0 | 0 | 0.1 | 0 |
| Gastropoda 3 (Gas3) | 0.3 | 0.1 | 0.1 | 0.6 |
| *Grandidierella japonica* (GRJA) (g, i) | 31.8 | 1.8 | 2.7 | 2.3 |
| Ostracoda (Ostr) | 0 | 0 | 0.2 | 0.1 |
| *Paradexamine* sp. (PAsp1) (g) | 0.3 | 1.7 | 7.3 | 6.1 |
| *Pentidotea resecata* (PERE) (I, n) | 1 | 11.5 | 0 | 0 |
| *Phyllaplysia taylori* (PHTA) (n) | 0 | 1.3 | 0 | 0 |
| Year 2 | | | | |
| Summer 2014 | | | | |
| *Ampelisca abdita* (AMAB) (g, i) | - | - | 0.1 | 0.1 |
| *Ampithoe valida* (AMVA) (g, i) | - | - | 33.1 | 33.6 |
| Annelida 1 (Ann1) | - | - | 0.3 | 6.3 |
| Annelida 2 (Ann2) | - | - | 3.8 | 2.5 |
| Bivalvia (Biva) | - | - | 0.3 | 0.2 |
| *Caprella* sp. (CAsp) (c) | - | - | 0.3 | 1.5 |
| Cirripedia (Cirr) | - | - | 1.8 | 1.3 |
| Corophiidae (Coro) (g) | - | - | 1049.5 | 471.9 |
| Gastropoda 1 (Gas1) | - | - | 0 | 0.2 |
| Gastropoda 3 (Gas3) | - | - | 0 | 0.5 |
| *Gemma gemma* (GEGE) (i) | - | - | 0.2 | 0.8 |
| *Grandidierella japonica* (GRJA) (g, i) | - | - | 2.1 | 0.1 |
| Isopoda (Isop) (I) | - | - | 0 | 0.1 |
| *Jassa slatteryi* (JASL) (g, c) | - | - | 75.1 | 8.8 |
| *Jassa* sp. (JAsp) (g) | - | - | 0.7 | 0 |
| Ostracoda (Ostr) | - | - | <0.1 | 1.6 |
| *Paradexamine* sp. (PAsp1) (g) | - | - | 0.9 | 1.9 |
| *Phyllaplysia taylori* (PHTA) (n) | - | - | 0.1 | 0 |
| Platyhelminthes (Plat) | - | - | 0 | <0.1 |
| *Potamocorbula amurensis* (POAM) (i) | - | - | 0 | <0.1 |
| *Siliqua patula* (SIPA) (n) | - | - | 0 | 0.8 |
| Tanaidacea (Tana) | - | - | <0.1 | 0.5 |

**Table A3.** *Cont.*

| Taxa | KB Natural | PM Natural | SR Eelgrass | SR Eelgrass + Oyster |
|---|---|---|---|---|
| **Fall 2014** | | | | |
| *Ampelisca abdita* (AMAB) (g, i) | 0 | 0 | 0.1 | 0.1 |
| Amphipoda (Amph) | 0 | 0 | 7.5 | 1 |
| *Ampithoe valida* (AMVA) (g, i) | 2.2 | 4.4 | 5.5 | 2.9 |
| Annelida 1 (Ann1) | 0 | 0 | 1.5 | 2.1 |
| Annelida 2 (Ann2) | 1 | 1.1 | 1.5 | 2.9 |
| Bivalvia (Biva) | 0 | 0 | <0.1 | 0.1 |
| *Caprella californica* (CACA) (c, n) | 0 | 0 | 0.3 | 0.2 |
| *Caprella* sp. (CAsp) (c) | 0 | 0 | 8.6 | 1.1 |
| Cirripedia (Cirr) | 0 | 0 | <0.1 | 0.5 |
| Corophiidae (Coro) (g) | 0.5 | 1.6 | 1 | 6.4 |
| Cumacea (Cuma) | 0 | 0 | 0 | <0.1 |
| *Gammarus daiberi* (GADA) (g, i) | 0 | 0 | 0 | 0.1 |
| Gastropoda 3 (Gas3) | 0 | 0 | 0.1 | 0 |
| *Gemma gemma* (GEGE) (i) | 0 | 0 | 0 | <0.1 |
| *Gnorimosphaeroma oregonensis* (GNOR) (I, n) | 0 | 0 | <0.1 | 0.1 |
| *Grandidierella japonica* (GRJA) (g, i) | 0 | 0 | 0 | 0.1 |
| Isopoda (Isop) (I) | 0 | 0 | 0 | 0 |
| *Jassa slatteryi* (JASL) (g, c) | 0 | 0.1 | 56.8 | 5.9 |
| *Jassa* sp. (JAsp) (g) | 0 | 0 | 0.1 | 0.1 |
| Ostracoda (Ostr) | 0 | 0.1 | 1 | 5 |
| *Paradexamine* sp. (PAsp1) (g) | 0 | 0 | 23.3 | 38 |
| *Paranthura* sp. (PAsp2) (i) | 0 | 0 | 0.5 | 0.5 |
| *Pentidotea resecata* (PERE) (I, n) | 0 | 0 | 0 | <0.1 |
| *Phyllaplysia taylori* (PHTA) (n) | 0 | 0.7 | 0 | 0 |
| Platyhelminthes (Plat) | 0 | 0 | <0.1 | 0 |
| *Stenothoe valida* (STVA) (g, i) | 0 | 0 | 20.9 | <0.1 |
| *Synidotea laticauda* (SYLA) (i, i) | 0 | 0 | 0.3 | 0.1 |
| Tanaidacea (Tana) | 0.2 | 1 | 0.2 | 0.2 |
| *Allorchestes angusta* (ALAN) (g, n) | 0 | 0 | 0.1 | 0.1 |
| Amphipoda (Amph) | 6.4 | 0.1 | 3.4 | 4.9 |
| *Ampithoe valida* (AMVA) (g, i) | 0.1 | 1.2 | 6.1 | 3.9 |
| Annelida 1 (Ann1) | 0.4 | 0 | 3.2 | 50.4 |
| Annelida 2 (Ann2) | 0.1 | 0.6 | 2.8 | 3.9 |
| Bivalvia (Biva) | 0 | 0 | 0.1 | 0.3 |
| *Caprella californica* (CACA) (c, n) | 1.6 | 3.2 | 2.7 | 0.7 |
| *Caprella drepanochir* (CADR) (c, i) | 5.9 | 0 | 0.1 | 0 |
| *Caprella* sp. (CAsp) (c) | 125.1 | 4.4 | 6.3 | 0.3 |
| Cirripedia (Cirr) | 0 | 0 | 0.4 | 0 |
| Corophiidae (Coro) (g) | 0 | 0 | 11.5 | 8.8 |
| Cumacea (Cuma) | 31.5 | 0 | 0.1 | 0.1 |
| *Gammarus daiberi* (GADA) (g, i) | 0 | 0 | 0 | 0.1 |
| Gastropoda 1 (Gas1) | 0 | 0 | 0.3 | 0 |
| Gastropoda 3 (Gas3) | 0 | 0 | 0.9 | 2.3 |
| *Gnorimosphaeroma oregonensis* (GNOR) (I, n) | 0 | 0 | 0.1 | 0 |
| *Jassa slatteryi* (JASL) (g, c) | 1.1 | 0.3 | 1 | 0.4 |
| Ostracoda (Ostr) | 0 | 0 | 14.5 | 23.4 |
| *Paradexamine* sp. (PAsp1) (g) | 0.3 | 0 | 0.2 | 0.5 |
| *Paranthura* sp. (PAsp2) (i) | 0 | 0 | 0 | 0.1 |
| *Pentidotea resecata* (PERE) (I, n) | 0.1 | 0 | 0 | 0 |
| *Phyllaplysia taylori* (PHTA) (n) | 0 | 1.1 | 0.2 | 0 |
| Platyhelminthes (Plat) | 0 | 2.2 | <0.1 | 0.9 |
| *Stenothoe valida* (STVA) (g, i) | 0 | 0 | 0.3 | 0.1 |
| Tanaidacea (Tana) | 0 | 0 | 0.3 | 1 |

**Table A3.** *Cont.*

| | KB | PM | SR | SR |
|---|---|---|---|---|
| **Taxa** | **Natural** | **Natural** | **Eelgrass** | **Eelgrass + Oyster** |
| Year 3 | | | | |
| Summer 2015 | | | | |
| *Allorchestes angusta* (ALAN) (g, n) | 0 | 0 | 0 | 0.1 |
| Amphipoda (Amph) | 6.2 | 1.3 | 1.1 | 1.3 |
| *Ampithoe lacertosa* (AMLA) (g, n) | 0.1 | 0 | 0.1 | 0.1 |
| *Ampithoe valida* (AMVA) (g, i) | 0.6 | 0.4 | 3.5 | 3.1 |
| Annelida 1 (Ann1) | 0.2 | 0 | 0.4 | 6.2 |
| Annelida 2 (Ann2) | 3.7 | 3.1 | 8.2 | 10.8 |
| *Caprella californica* (CACA) (c, n) | 0.5 | 0.8 | 1.6 | 1.6 |
| *Caprella drepanochir* (CADR) (c, i) | 1.3 | 0 | 0 | 0.1 |
| *Caprella* sp. (CAsp) (c) | 3.1 | 0 | 0.1 | 0.8 |
| Cirripedia (Cirr) | 0 | 0 | 0.1 | 0 |
| Corophiidae (Coro) (g) | 1.5 | 1.4 | 5.8 | 2.9 |
| Cumacea (Cuma) | 0 | 0 | 0.5 | 0.1 |
| Gastropoda 3 (Gas3) | 0.5 | 0 | 2 | 2.1 |
| *Gemma gemma* (GEGE) (i) | 0 | 0.1 | 0 | 0 |
| *Grandidierella japonica* (GRJA) (g, i) | 0 | 0 | 0 | 0.2 |
| *Jassa slatteryi* (JASL) (g, c) | 13.4 | 5.2 | 0.1 | 0 |
| Ostracoda (Ostr) | 0.2 | 0.5 | 3.1 | 10.6 |
| Paradexamine sp. (PAsp1) (g) | 1.5 | 0.7 | 1.6 | 0.7 |
| Paranthura sp. (PAsp2) (i) | 0 | 0.2 | 1.5 | 0.7 |
| *Pentidotea resecata* (PERE) (I, n) | 0.2 | 1.4 | 0 | 0 |
| *Potamocorbula amurensis* (POAM) (i) | 0 | 0 | 0.1 | 0 |
| *Synidotea laticauda* (SYLA) (I, i) | 0 | 0 | 0.1 | 0 |
| Tanaidacea (Tana) | 0.1 | 0 | 0 | 0 |

**Table A4.** perMANOVA based on Bray–Curtis dissimilarity metric on abundance of taxa collected during suction sampling at San Rafael.

| Factor | Class [1] | Pr(>F) | $r^2$ |
|---|---|---|---|
| | **Years 0 and 1 [2]** | | |
| Treatment | P, E, EO(E), O, EO(O), C | <0.001 | 0.25 |
| Season | Fall, Spring, Summer, Winter | <0.001 | 0.15 |
| Treatment × Season | | <0.001 | 0.23 |
| | **Years 0, 1, 2, and 3 [3]** | | |
| Treatment | P, E, EO(E), O, EO(O), C | <0.001 | 0.21 |
| Season | Fall, Spring, Summer, Winter | <0.001 | 0.11 |
| Treatment × Season | | <0.001 | 0.20 |
| | **Years 0, 1, 2, and 3 [4]** | | |
| Treatment | P, E, EO(E), O, EO(O), C | <0.001 | 0.26 |
| Year | 0, 1, 2, and 3 | <0.001 | 0.34 |
| Treatment × Year | | <0.001 | 0.16 |

[1] Classes: C = control, E = eelgrass, EO(O) = oyster within combination eelgrass and oyster, EO(E) = eelgrass within combination eelgrass and oyster, O = oyster, and P = pre-treatment. [2] Pre-treatment surveys (Year 0) conducted October 2011–July 2012 and Year 1 treatment surveys conducted July 2013–April 2014. [3] Pre-treatment surveys (Year 0) conducted October 2011–July 2012 and treatment surveys conducted July 2013–August 2015 (Years 1–3). [4] Summer pre-treatment survey conducted July 2012 (Year 0) and summer treatment surveys conducted July 2013 (Year 1), July 2014 (Year 2), and July 2015 (Year 3).

**Table A5.** perMANOVA based on Bray–Curtis dissimilarity metric on abundance of taxa collected during shoot sampling at San Rafael, Point Molate, and Keller Beach.

| Factor | Class [1] | Pr(>F) | $r^2$ |
|---|---|---|---|
| Treatment | E, EO, N | <0.001 | 0.07 |
| Source Site | SR(E), SR(EO), PM, KB | <0.001 | 0.10 |
| Year | 1,2,3 | <0.001 | 0.10 |
| Treatment × Year | | <0.001 | 0.08 |
| Source Site × Year | | <0.001 | 0.10 |

[1] Classes: E = eelgrass restored, EO = eelgrass + oyster combination restored, N = natural site; SR = San Rafael, PM = Point Molate, KB = Keller Beach; Year 1 (summer 2013, fall 2013, spring 2014), Year 2 (summer 2014, fall 2014, spring 2015), and Year 3 (summer 2015).

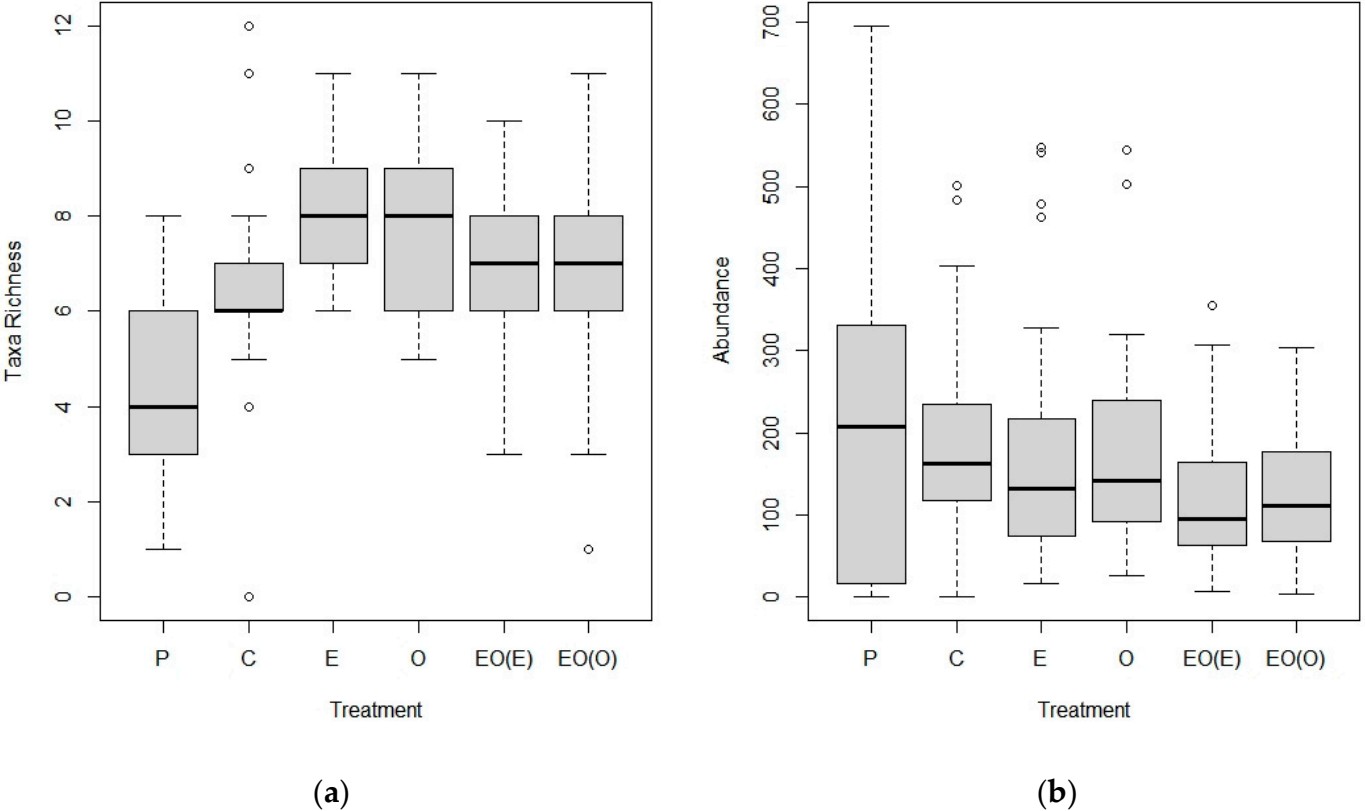

(**a**)                                      (**b**)

**Figure A1.** Taxa richness (**a**) and total abundance (**b**) of suction-sampled invertebrates per sample, pre-treatment (four sampling periods pooled for Oct 2011–July 2012 for n = 24 samples) and across all quarterly post-installation sampling periods: (six sampling periods pooled for Aug 2013-Aug 2015, for n = 30 samples per treatment). Treatments included: P = pre-treatment, C = control, E = eelgrass, O = oyster, EO(E) = eelgrass from combination plots, EO(O) = oyster from combination plots. Upper whisker is 3rd quartile (Q3) ×1.5 × inner quartile range (IQR) and the lower whisker is 1st quartile (Q1) ×1.5 × IQR; dots are data points outside the whiskers.

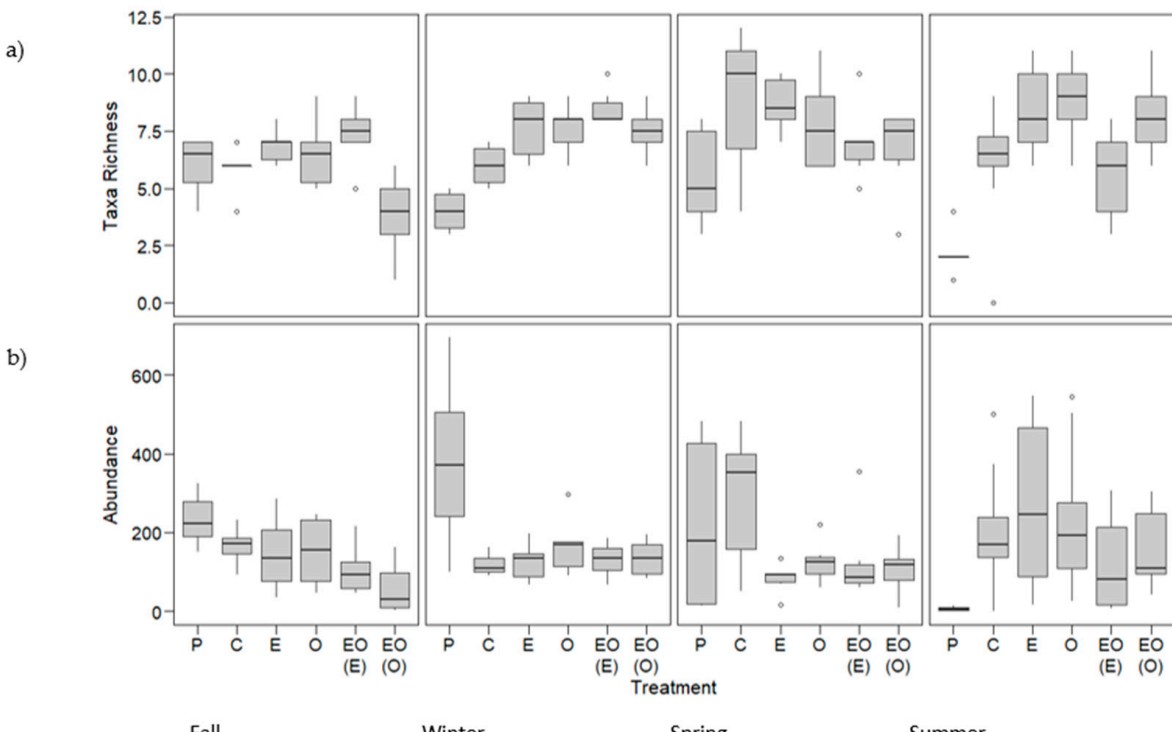

**Figure A2.** Taxa richness (**a**) and total abundance (**b**) of suction-sampled invertebrates by season. Each season contains one pre-treatment sampling period compared to a corresponding post-treatment period (for fall, pre was 2011 and post was 2014; for winter and spring, pre was 2012 and post was 2014; for summer, pre was 2012 and post was 2013, 2014, and 2015). Treatments included: P = pre-treatment, C = control, E = eelgrass, O = oyster, EO(E) = eelgrass from combination plots, EO(O) = oyster from combination plots. Boxplot symbols as in Appendix A Figure A1.

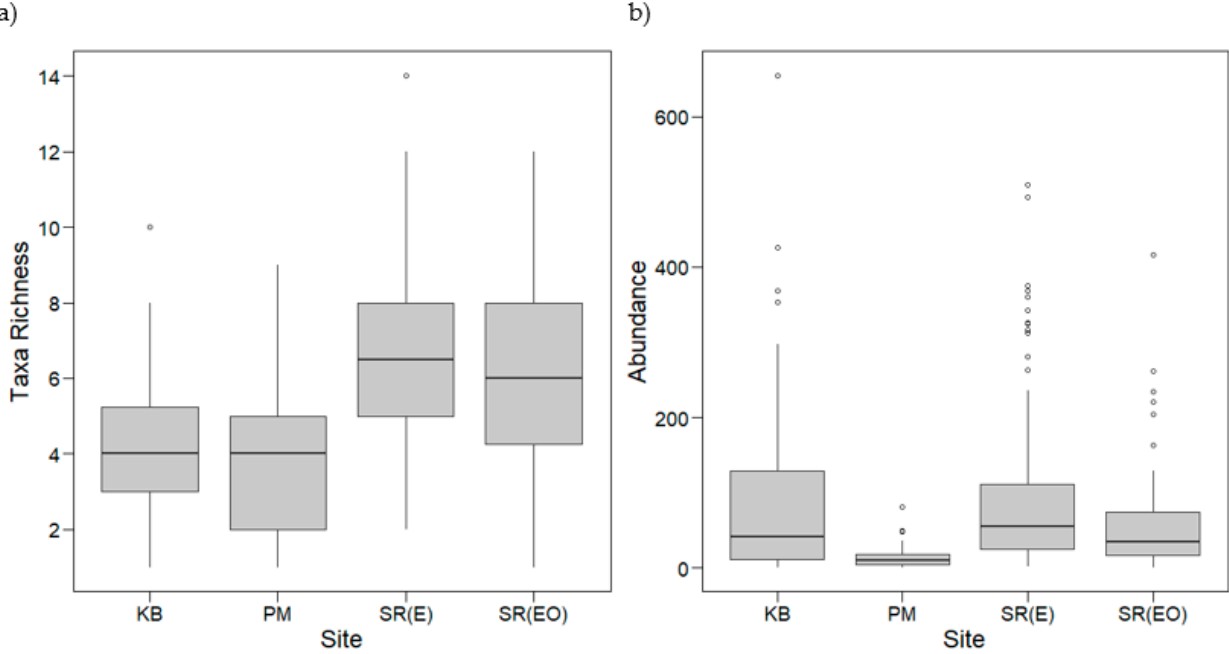

**Figure A3.** Taxa richness (**a**) and abundance (**b**) of invertebrates from shoot samples collected at natural eelgrass beds Keller Beach (KB, n = 36) and Point Molate (PM, n = 38), and restored populations at the San Rafael project site (SR, n = 102) across three years of post-treatment sampling of eelgrass. Boxplot symbols as in Appendix A Figure A1.

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
