# Peer review of "Seagrass and Oyster Reef Restoration in Living Shorelines: Effects of Habitat Configuration on Invertebrate Community Assembly"

_diversity, doi:10.3390/d13060246_

Round 1

Reviewer 1 Report

The study by Pinnel et al. is an interesting MS, as it compares the restoration of eelgrass and oyster reef meadows in the SF estuary with an enormous quantity of work done. Starting with the positive part, it is observed rapidly (within a year) changes in the diversity of the invertebrate community in response to restoration, as well as the interaction between eelgrass and oyster reef. Statistical methods seem appropriate, and changes in correspondence analysis between treatments, as well as seasonal changes are clearly detected. However, there are a number of points that I would like the authors to answer appropriately and consider before accepting the MS.

In the first place, work is done on a plot in San Rafael, but nothing is mentioned about the previous presence of both types of communities in this place. There is no indication beyond the suitability of the site in terms of depth and type of substrate, as well as previous pilot transplant trials. My question is, did eelgrass exist in the past on this site? Are there records about that?.

Second, differences in specific composition are observed between restored eelgrass meadows and natural control populations at Keller Beach and Point Mollate. Reading the results (point 3.2.2.), I wonder to what extent the habitats are similar, especially in terms of type of substrate and hydrodynamics. Later the authors agree with me in discussion, lines 380s, as they stated differences between the two points in addition to the restored plots, and argue about the sediment or water conditions. Are there data from other publications that characterize the habitat of these places from a physical-chemical point of view? The MS would improve with this information. The fact that the control sites have different characteristics makes comparison difficult.

Third, the influence of the closeness of the interspersed plots on the abundance and specific richness between the E and OE is observed. Although this may be interesting from the restoration point of view, it undoubtedly makes it difficult to interpret the results as independent treatments, since they are very close to the possible flows of individuals among them. That calls into question a part of the conclusions of the work. If the restoration has worked, there is a lack of sample independence among treatments.

Finally, the fact that exotic species are one of the first arriving to restored habitats raises an interesting question about how to manage it that must be deepen in the text. How to avoid that exotic species will be favoured by habitat restoration.

Minor points.

Line 85. 8,000 acres, please use Ha. (International system units). 3238Ha.

Line 176. It is necessary to specify the type (brand?) of mesh?. It may be sufficient with nylon mesh,

Line 198. Invertebrate samples from natural sites were stained with rose Bengal dye ?. As in the case of the samples of lines 183-184 ?.

I hope all these comemts will help authors to improve the MS.

Reviewer 2 Report

Review

Paper title: Seagrass and oyster reef restoration in living shorelines: effects of habitat configuration on invertebrate community assembly

San Francisco Bay is a large estuary in Northern California, USA, which is surrounded by a major urban area. This region has lost nearly 80% of its historic tidal marsh habitat over the last 150 years. In this area, there has been increased focus on coastal margin habitat restoration over the last decade to improve habitat conditions of the Olympia oyster Ostrea lurida. In this paper, the authors provided the current results of the San Francisco Bay Subtidal Habitat Goals Project. They found that oyster reefs and eelgrass supported unique invertebrate communities, taxa richness increased between the pre-treatment and post-treatment period, but invertebrate abundances were similar. According to shoot collections, community assemblages of restored and natural eelgrass were not equivalent, because some species had lower abundance.

These results are important in terms of ecological monitoring, habitat restoration patterns, and coastal engineering. 

All these reasons explain the relevance of the paper by Cassie M. Pinnell and co-authors submitted to "Diversity".

General scores.

The data presented by the authors are original and significant. All conclusions are justified and supported by the results. The study is correctly designed and technically sounds. In general, the statistical analyses are performed with good technical standards. We authors conducted careful work which will attract the attention of a wide range of specialists focused on the benthic ecology, restoration, coastal managers and ecologists.

Specific comments.

L 64. Change “virginica,” to “virginica

L 85. Please, re-calculate 8,000 acres to square meters and provide this value here.

L 137. Change “Combined oyster/eelgrass” to “Oyster/eelgrass”

L 179. "In the lab”. Consider adding a comma.

L 223. Change “signficant” to “significant”

L 260. Change “illustrates” to “illustrate”

L 285. Change “signficant” to “significant”

L 288. Change “signficanlty” to “significantly”

L 290. Change “signficanlty” to “significantly”

L 291. Change “signficant” to “significant”

L 402-413 should be formatted as "Conclusion".

The authors assessed the efficacy of restored intertidal eelgrass beds and native Olympia oyster reefs alone and together in terms of assemblage structure and species abundance. However, the production and biomass of new communities are important indicators in such studies. I recommend the authors include the data on how the community biomass changed across their study (at least visual observations).

References should be formatted according to "Instructions for authors": remove "vol.", "no." "pp.", use short Journal titles (for example "Mar. Ecol. Prog. Ser." instead of "Marine Ecology Progress Series").

Reference 10. " Zostera marina" and " Ascophyllum nodosum" should be italicized

Reference 13. " Zostera marina" and "Crassostrea gigas" should be italicized

Reference 15. " Zostera marina" should be italicized

Reference 20. " Ostrea lurida" should be italicized

Reference 25. " Zostera marina" should be italicized

Reference 31. " Zostera marina" should be italicized

Reference 33. " Parophrys vetulus" and " Cancer magister" should be italicized

Reference 34. " Cancer magister" should be italicized

Reference 35. " Cancer magister" should be italicized

Reference 37. Change “Zosetera” to “Zostera”, "Zostera marina" should be italicized

Round 2

Reviewer 1 Report

The authors have adequately responded to the points raised in the first review. Therefore, my assessment is to accept the MS with minor changes, basically editorial, arising in preparation of the MS for publication.

Best regards